

# Two membrane-bound transcription factors regulate expression of various type-IV-pili surface structures in *Sulfolobus acidocaldarius*

Lisa Franziska Bischof[1,2], Maria Florencia Haurat[3] and Sonja-Verena Albers[1]

[1] Institute of Biology II, Molecular Biology of Archaea, University of Freiburg, Freiburg, Germany
[2] Spemann Graduate School of Biology and Medicine (SGBM), Freiburg, Germany
[3] Department of Molecular Microbiology, Washington University, School of Medicine in St. Louis, St. Louis, MO, USA

Corresponding author
Sonja-Verena Albers,
sonja.albers@biologie.uni-freiburg.de

## ABSTRACT

In Archaea and Bacteria, gene expression is tightly regulated in response to environmental stimuli. In the thermoacidophilic crenarchaeon *Sulfolobus acidocaldarius* nutrient limitation induces expression of the archaellum, the archaeal motility structure. This expression is orchestrated by a complex hierarchical network of positive and negative regulators—the archaellum regulatory network (arn). The membrane-bound one-component system ArnR and its paralog ArnR1 were recently described as main activators of archaellum expression in *S. acidocaldarius*. They regulate gene expression of the archaellum operon by targeting the promoter of *flaB*, encoding the archaellum filament protein. Here we describe a strategy for the isolation and biochemical characterization of these two archaellum regulators. Both regulators are capable of forming oligomers and are phosphorylated by the Ser/Thr kinase ArnC. Apart from binding to *pflaB*, ArnR but not ArnR1 bound to promoter sequences of *aapF* and *upsX*, which encode components of the archaeal adhesive pilus and UV-inducible pili system, demonstrating a regulatory connection between different surface appendages of *S. acidocaldarius*.

## INTRODUCTION

Stress response that yields adaptation to changing environmental conditions is one of the most important prerequisites to ensure survival in prokaryotes. Various signal transduction modules have evolved to receive, transfer and process extracellular signals in the cell (*Esser et al., 2016*; *Koretke et al., 2000*; *Ulrich, Koonin & Zhulin, 2005*). In response to these signals, transcription factors regulate and adjust gene expression to ensure cellular survival. In Bacteria, a variety of alternative sigma factors recognize and bind promoter sequences in response to changing environmental conditions and target the

transcription machinery towards genes whose products are required to survive the given circumstances (*Paget, 2015*). Several helix-turn-helix (HTH) domain containing transcription factors were described in Archaea as transcription regulators of e.g. the central carbon- and energy- as well as amino acid metabolism (*Peeters & Charlier, 2010*). The crenarchaeon *Sulfolobus acidocaldarius* produces three cell surface appendages that are homologous to bacterial type-IV-pili (T4P). In archaea, T4P have functions such as surface attachment, biofilm formation, cell aggregation and motility (*Albers & Meyer, 2011*; *Albers & Pohlschröder, 2009*; *Makarova, Koonin & Albers, 2016*; *Pohlschroder et al., 2011*). The archaeal adhesive pilus (Aap)-pilus is the most abundant cell surface structure in exponentially growing *S. acidocaldarius*. Aap-pili are required for adhesion to surfaces and are involved in biofilm formation (*Henche et al., 2012b*, *2012a*). Other surface appendages are produced in response to environmental changes, e.g. the UV-inducible pili system (Ups), which is induced upon UV-irradiation and other stress factors that promote DNA-double strand breaks. Ups-pili are used to form cellular aggregates that allow DNA exchange between cells via the Ced System. *S. acidocaldarius* can thereby repair (UV-induced) double strand breaks via homologous recombination (*Fröls et al., 2008*; *van Wolferen et al., 2013*; *van Wolferen et al., 2016*). Apart from that, *S. acidocaldarius* produces its motility structure, the archaellum, in response to nutrient limitation (*Lassak et al., 2012*). The archaellum of *S. acidocaldarius* consists of seven proteins (FlaB, FlaX, FlaG, FlaF, FlaH, FlaI and FlaJ) that are encoded in an operon whose transcription is controlled by two promoters (p*flaB* and p*flaX*) (*Lassak et al., 2012*). Expression of the archaellum is tightly regulated by a complex network of positive and negative regulators and phosphorylation by various protein kinases plays a fundamental role in the regulatory process (*Hoffmann et al., 2016*; *Haurat et al., 2017*; *Li et al., 2017*). Especially the promoter upstream of the gene encoding the archaellum filament protein FlaB is strongly induced upon nutrient depletion (*Lassak et al., 2012*, *2013*). Under nutrient limiting conditions, the membrane-bound transcription regulator ArnR is required for the induction of archaellum expression. While ArnR is conserved in Sulfolobales and Desulfurococcales, the ArnR paralog ArnR1 is exclusively found in *S. acidocaldarius* (*Lassak et al., 2013*). ArnR and ArnR1 are two one-component systems—the predominant class of regulatory systems in archaea and bacteria (*Ulrich, Koonin & Zhulin, 2005*). They encompass an almost identical N-terminal HTH domain; a putative HAMP (present in histidine kinases, adenyl cyclases, methyl-accepting proteins and phosphatases) and sensing domain presumably involved in sensing and transducing a starvation-related signal; and two C-terminal transmembrane domains (Fig. 1) (*Lassak et al., 2013*). The deletion of either of the two regulators leads to reduced cell motility under nutrient depletion, with the deletion of *arnR* causing a stronger reduction than the deletion of *arnR1* (*Lassak et al., 2013*). The simultaneous deletion of both regulators leads to non-motile cells (*Lassak et al., 2013*). Conditions under which expression of *arnR1* is induced are unknown, whereas transcription of *arnR* increases upon nutrient limitation. Both proteins presumably target the *flaB* promoter, which harbors two conserved cis-regulatory elements called ArnR
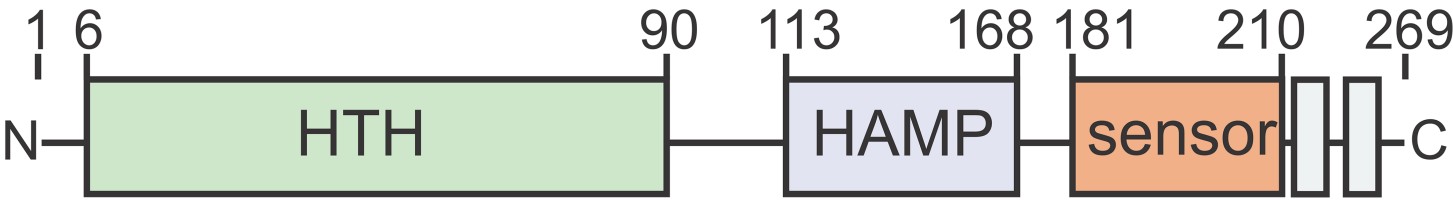

**Figure 1 ArnR and ArnR1 share an overall domain organization.** Both proteins harbor an N-terminal helix-turn-helix (HTH, green)) domain, a putative HAMP (histidine kinases, adenyl cyclases, methyl-accepting proteins and phosphatases, blue) domain, a sensory domain (orange) and two C-terminal transmembrane domains (gray). Numbers correspond to the first and last amino acid of each domain (*Lassak et al., 2013*).

box-1 and -2. It was shown that deletion of both boxes abolishes and mutation of box-2 strongly decreases *flaB* promoter activity (*Lassak et al., 2013*).

Recently, the deletion of the core component of the Aap was found to result in upregulation of the archaellum operon and hyperarchaellated cells (*Henche et al., 2012a*). Based on these findings, cross-regulation between the archaellum and Aap-pili was proposed, but the underlying system was not identified so far. Here, we set out to study the regulatory function of the two one-component systems ArnR and ArnR1 on T4P surface structures of *S. acidocaldarius*. Purification strategies for both membrane proteins in full-length were developed and their oligomeric state was assessed. We analyzed if ArnR and ArnR1 are targets of the eukaryotic-like protein kinases ArnC and ArnD, which are well-known archaellum regulators (*Hoffmann et al., 2016*). Lastly, an in vitro assay was established to assess their binding affinities to different T4P promoters and qRT-PCR was performed to confirm the results in vivo. To our knowledge, this is the first study of archaeal membrane-bound transcription regulators.

## MATERIALS AND METHODS

### *Sulfolobus acidocaldarius* strains, plasmids and growth conditions

All strains used in this study are described in Table S1. Strains were grown essentially as described, using basal Brock medium (pH 3.5) supplemented with 0.1% NZ-amine, 0.2% sucrose and 10 μg/ml Uracil (*Wagner et al., 2012*). Plasmids and their creation are described in Table S2. Primers are listed in Table S3.

### *Escherichia coli* strains, plasmids and growth conditions

All *E. coli* strains used in this study are described in Table S1. All strains were grown in LB medium supplemented with the respective antibiotics. Plasmids were generated as described in Table S2 using primers described in Table S3.

### SDS-PAGE and Western-blot analysis

Total membranes or purified protein samples were supplemented with 5× SDS-loading buffer (10% (w/v) SDS, 300 mM Tris/HCl pH 6.8, 500 mM DTT, 50% (v/v) Glycerol, 0.04% (w/v) bromphenole blue) and subjected to SDS-PAGE analysis according to the method of Laemmli using 11% gels. For Western-blot analysis, proteins were

transferred onto PVDF membranes (Roche diagnostics) using the semi-dry method. His-HRP antibody (Abcam) diluted 1:10,000 in PBST was used to detect ArnR and ArnR1. PBST was prepared by diluting a $10 \times$ PBS stock (composition: 580 mMNa$_2$HPO$_4$ $\times$ 2H$_2$O, 170 mM NaH$_2$Po$_4$ $\times$ H$_2$O, 680 mM NaCl, 0.05% Tween20 pH 7.3) to $1 \times$ PBS using distilled H$_2$O and 0.05% Tween20 was added. Chemiluminescent signals were detected as described (*Hoffmann et al., 2016*).

## Expression and purification of ArnR and ArnR1

Overexpression of ArnR from pSVA2543 and ArnR1 from pSVA2538 was performed essentially as described (*Studier, 2014*) using the *E. coli* BL21 (DE3) derivative strain *E. coli* OverExpress(tm)C43(DE3) (Lucigen). Cells were grown for approximately 48 h to OD$_{600}$ of 12. Membranes were isolated essentially as described (*Bischof et al., 2016*), using buffer A (20 mM Tris/HCl, pH 8, 300 mM NaCl). solubilization was performed for 2 h at 4 °C, using five mg/ml total membranes in solubilization buffer (buffer A supplemented with 10 mM Imidazole, 2% *n*-Dodecyl-β-d-maltopyranoside (DDM) (Glycon)). Solubilized protein was isolated from residual membranes using ultracentrifugation (400,000 $\times$ *g*, 4 °C, 1 h) and purified using one ml of His-Select nickel affinity gel resin (Sigma) equilibrated in solubilization buffer. Solubilized protein was applied to the resin and the flow-through was collected. After washing of the column with $5 \times 1$ ml of wash buffer (buffer A supplemented with 20 mM Imidazol, 0.5% DDM (Glycon)), bound target protein was eluted using $5 \times 500$ µl buffer A supplemented with 0.03% DDM and 150 mM Imidazole. Eluted ArnR protein was desalted to buffer A supplemented with 0.03% DDM using a PD10 column (GE Healthcare, Chicago, IL, USA), according to the manufacturer's protocol. Eluted ArnR1 protein was further purified using a Heparin-HP column (GE Healthcare, Chicago, IL, USA) on an Äkta purifier system (GE Healthcare, Chicago, IL, USA). Therefore, fractions containing ArnR1 were diluted to 20 mM NaCl in buffer A supplemented with 0.03% DDM and loaded on a five ml HisTrap HP at one ml/min flow rate. After washing of the column with 20 mM Tris/HCl pH 8, 20 mM NaCl, 0.03% DDM for at least five column volumes, ArnR1 was eluted in two ml fractions using 20 mM Tris/HCl, pH 8, 500 mM NaCl, 0.03% DDM. ArnR as well as ArnR1 was concentrated using ultrafiltration in a 100 kDa cut-off Amicon (Merck Millipore, Burlington, MA, USA). Protein concentration was determined using BCA assay (Serva), according to the manufacturer's protocol. The Analysis of the oligomeric state of purified ArnR and ArnR1 was performed using Blue-Native PAGE analyses as described (*Claeys, Geering & Meyer, 2005*) using NativeMark$^{TM}$ unstained protein standards (Life Technologies, Inc., Carlsbad, CA, USA).

### In vitro phosphorylation assays

In vitro phosphorylation assays of ArnR and ArnR1 with γ[$^{32}$P]-ATP (Hartmann Analytic) were performed as described (20). The reaction buffer contained 20 mM Tris/HCl pH 8, 150 mM NaCl. In a total final volume of 15 µl, two µM kinase and three µM ArnR or ArnR1 were mixed and 0.8 mM non-radioactive ATP and 0.3 mM γ[$^{32}$P]-ATP were added to the samples.

To show that phosphorylation of the kinases was specific and further show that ArnR and ArnR1 do not possess auto-phosphorylation activity negative control samples were taken along, that contained only the respective protein and the ATP mixture. In another negative control, the kinases were incubated in the absence of ATP and ArnR and ArnR1. After incubation of all samples at 55 °C for 10 min, 5× SDS-loading dye was added to a final concentration of 1 times to stop the reaction. Proteins were separated on 11% SDS gels and exposed on a phosphostorage screen (Molecular Dynamics, Sunnyvale, CA, USA) overnight. Screens were scanned using Typhoon FLA 7000 (GE Healthcare, Chicago, IL, USA). Thee independent experiments were performed and a representative phosphoimage is shown.

## Microscale thermophoresis (MST)

For MST measurements, double-stranded AlexaFluor647-labeled DNA was generated by annealing two primers (Table S3). Therefore, primers were mixed in equal amounts (five mM final concentration) in 20 mM Tris/HCl pH 8, 200 mM NaCl and 0.05% (w/v) DDM, heated to 90 °C and then cooled to 20 °C in steps of 2 °C/10 s and annealing was monitored by analyzing primers before and after annealing on 15% TBE Gels. To determine the binding affinity of ArnR, 2.5 nM labeled, double-stranded *DNA* was titrated with increasing concentrations of ArnR (0.156–18 µM). The measurements were performed at 24 °C and 80% LED power. Samples were incubated for 10 min before measurements and centrifuged at 16,000 rpm in a tabletop centrifuge (Eppendorf, Hamburg, Germany). The laser on and off times were adjusted to 25 s and 5 s, respectively. Measurements were performed on a NanoTemper Monolith NT.115 instrument in hydrophilic-treated capillaries, and analyzed using NT analysis software version 1.4.27 (NanoTemper Technologies GmbH, Munich, Germany). The data of two independent experiments performed in duplicate were used to calculate binding affinities. Therefore, fluorescence was normalized ($\Delta F$) by subtracting the lowest measured fluorescence value determined within each experiment from all measured Fluorescence values. Obtained curves were fitted using non-linear regression, with one site-specific binding according to ($y = F_{max}*X/(K_D + X)$).

## Quantitative reverse transcription-polymerase chain reaction (qRT-PCR)

Total RNA samples were isolated from shaking cultures before (0 h) and after 2 h of starvation. Samples were isolated as described (*Lassak et al., 2012*). TRIzol reagent (Invitrogen, Carlsbad, CA, USA) was used for total RNA isolation following manufacturer's instructions. Digestion of gDNA and preparation of cDNA was performed using Quantinova Reverse Transcription Kit (Qiagen, Hilden, Germany), according to the manufacturer's protocol. The gDNA removal was performed for 3 min, and the Reverse Transcription reaction for 7 min. Quantitative reverse transcriptase (qRT-PCR) analysis of *aapF and upsX* was performed as described (*Haurat et al., 2017*) using Magnetic Induction Cycler (Bio Molecular Systems, Upper Coomera, QLD, Australia), 2× qPCRBIOSyGreen Mix Lo-ROX (Nippon genetics, Dueren, Germany) according to the

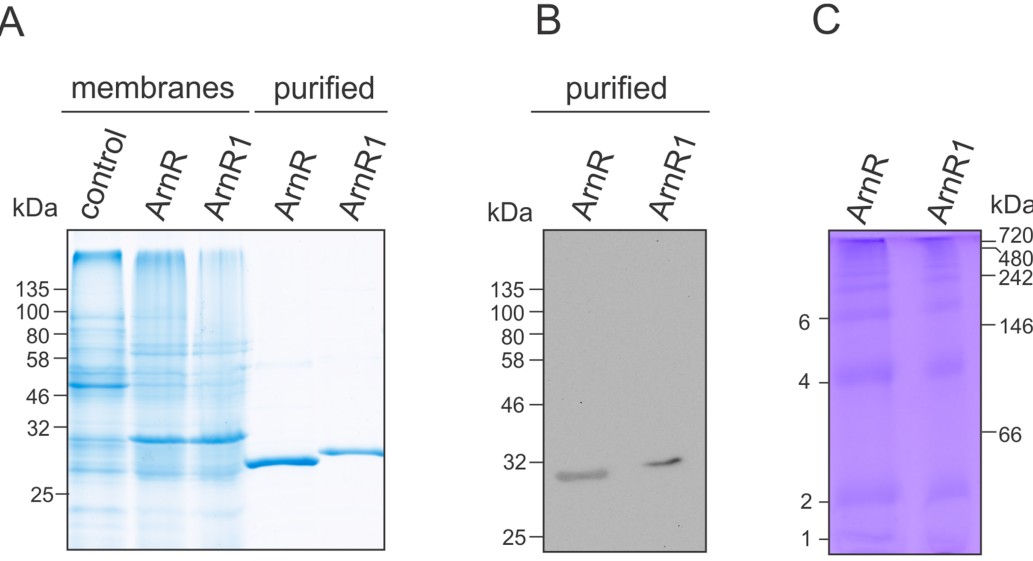

**Figure 2 Purification of codon optimized ArnR and ArnR1 and formation of diverse multimeric species.** (A) ArnR and ArnR1 were expressed in *E.coli* OverExpress C43(DE3). Membranes were isolated, and ArnR and ArnR1 were purified to homogeneity (A). Protein identity was confirmed using Western-blot analysis using His-specific antibodies (B). Numbers represent masses of marker proteins in kDa. Representative images of at least five purifications are shown. (C) Oligomerization of both regulators was analyzed using Blue-Native PAGE. Numbers on the right side represent mass of marker proteins in kDa. Various multimeric species were formed by ArnR and ArnR1, as estimated by numbers on the left. Three independent experiments were performed and a representative image is shown.

manufacturer's protocol, in combination with primers listed in Table S3. Three biological replicates and two technical replicates were assayed. Gene expression of *aapF* and *upsX* was normalized to their expression at time point 0 h (before starvation) and depicted in log2-fold change. Alcohol dehydrogenase (*saci_1690*) was used as housekeeping gene as transcript levels of this gene are not affected by nutrient limiting conditions (*Bischof et al., 2019*).

## RESULTS

### ArnR and ArnR1 form oligomeric structures

To characterize ArnR and ArnR1, both regulators were heterologously expressed in *E. coli* and purified to homogeneity (Fig. 2). To optimize translation rates for the respective proteins, ArnR and ArnR1 encoding genes were codon optimized for *E. coli* (Table S4) and synthetically manufactured (Genscript, Piscataway, NJ, USA). Codon optimized *arnR* and *arnR1* were expressed with an N-terminal His10-tag under control of an arabinose inducible promoter (*Geertsma & Dutzler, 2011*; *Valla & Lale, 2014*) in *E. coli* OverExpress C43(DE3) cells using autoinduction medium (*Studier, 2014*).

ArnR and ArnR1 were solely detected in the *E. coli* membrane fraction (Fig. 2A), which was used further for protein purification via Nickel-affinity chromatography coupled to desalting of eluted protein samples (Fig. 2A, middle). The identity of ArnR and ArnR1 was confirmed by immunoblot analysis using His-specific antibodies (Fig. 2B). To determine the oligomeric state of purified ArnR and ArnR1, Blue-Native PAGE

analysis was performed (Fig. 2C). ArnR and ArnR1 formed diverse multimeric species from monomeric to dodecameric (Fig. 2C). This is in agreement with the observation that transcription factors adopt homo-dimeric or multimeric states in their active conformation (*Wolberger, 1999*). The formation of various oligomeric species moreover showed that both regulators are stable and correctly folded after purification.

## ArnR and ArnR1 are phosphorylated by the Serine/Threonine kinase ArnC

Expression and presumably activity of the archaellum are regulated by reversible protein phosphorylation. The two eukaryotic protein kinases ArnC and ArnD phosphorylate multiple serine and threonine residues of the archaellum repressor ArnB (*Hoffmann et al., 2016*). ArnC (but not ArnD) further phosphorylates ArnA, which is thought to be essential for the interaction and activity of ArnA and ArnB (*Reimann et al., 2012*; *Hoffmann et al., 2016*). Phosphoproteome analysis revealed phosphorylation of Tyr217 and Thr222 in *S. acidocaldarius* ArnR1 and two neighboring tyrosine residues in the HAMP-domain (Tyr154 and Tyr155) of *S. solfataricus* ArnR (*Esser et al., 2012*; *Reimann et al., 2013*) (Sso_ArnR, Fig. 3A). Therefore, phosphorylation assays were performed next to analyze if ArnR and ArnR1 are phosphorylated in vitro. Given the fact that the protein kinases ArnC and ArnD are involved in archaellum regulation (*Hoffmann et al., 2016*), they were incubated with ArnR or ArnR1 in the presence of radioactively labeled $\gamma^{32}$P-ATP and phosphotransfer from the kinases to the regulators was monitored. Signals corresponding to auto-phosphorylation of ArnC (Fig. 3B, lane 4) and ArnD (Fig. 3C, lane 4) was obtained. Strikingly, phosphotransfer was observed from ArnC to both ArnR and ArnR1 (Fig. 3B, second to last and last lane), while no phosphotransfer to ArnR or ArnR1 was observed from ArnD (Fig. 3C, second to last and last lane). In addition, two distinct bands were obtained for phosphorylated ArnR and ArnR1 that might correspond to phosphorylated monomeric and dimeric species of the proteins (Fig. 3B, second to last and last lane), indicating that the multimerisation observed on BN-PAGE (Fig. 2B) might have a functional relevance.

## ArnR and ArnR1 bind to the *flaB* promoter

To understand their function as transcriptional activators of the archaellum operon, binding of full-length ArnR and ArnR1 to different *flaB* promoter sequences was investigated. It was recently described that the *flaB* promoter (p*flaB*) harbors two 15 bp cis-regulatory elements (ArnR box-1 and ArnR box-2), which are inverted repeats with the consensus sequence TCGAC-(N)5-GTCGA (*Lassak et al., 2013*) (Fig. 4A).

To characterize the binding of ArnR and ArnR1 to the *flaB* promoter, three different Alexa Fluor 647 labeled promoter fragments were generated and subjected to MST experiments in the presence of increasing amounts of detergent-purified ArnR or ArnR1. A 41 bp fragment containing both ArnR boxes (p*flaB*$^{41}$) as well as a fragment containing either ArnR box-1 or ArnR box-2 were used (Fig. 4A). With increasing concentrations of ArnR (Fig. 4B, left graph) or ArnR1 (Fig. 4B, right graph), binding to p*flaB*$^{41}$ was detected as was visualized by increasing fluorescence signals (Kd ArnR: 4.6μM, Kd ArnR1: 3.31 μM)

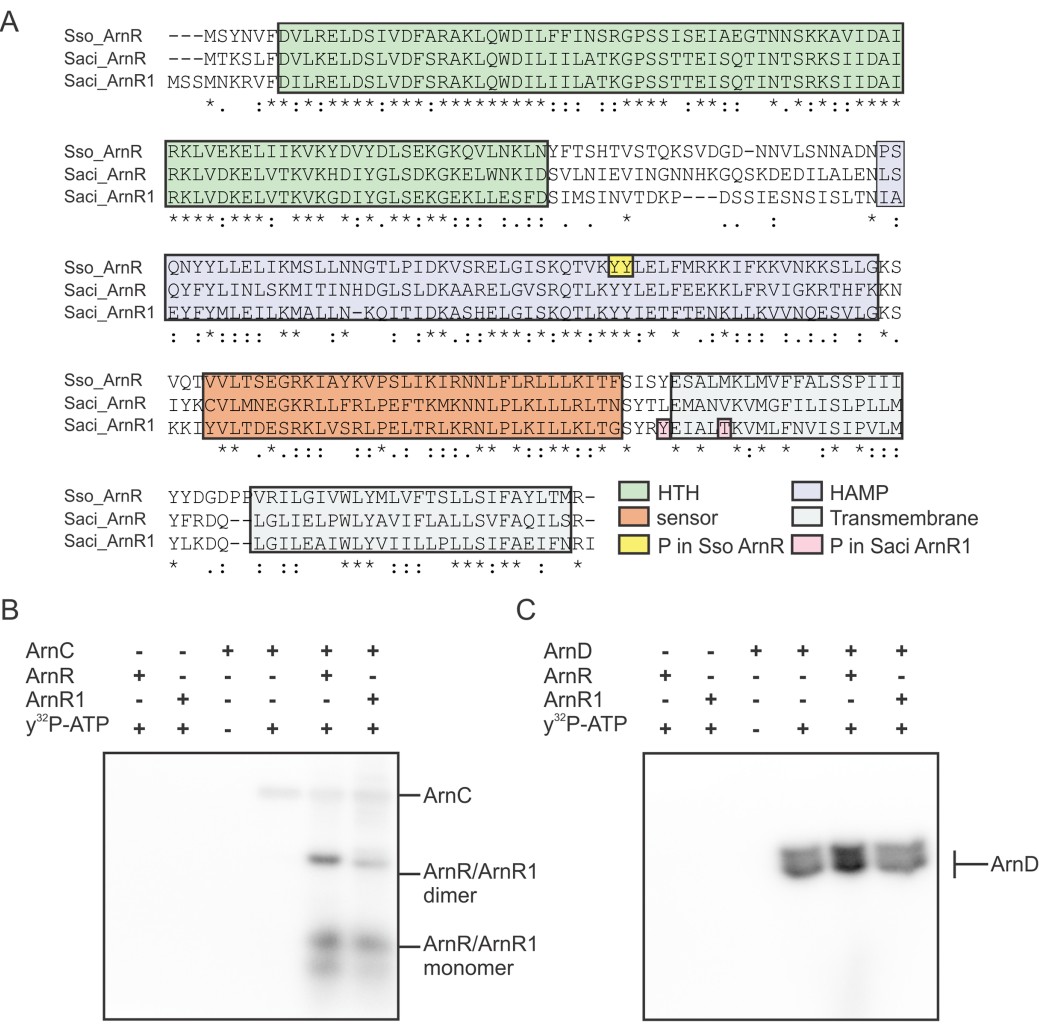

**Figure 3 ArnR and ArnR1 are phosphorylated by ArnC.** (A) Sequence alignment of *S. solfataricus* (Sso) ArnR and *S. acidocaldarius* (Saci) ArnR and ArnR1. Location of HTH (Helix-turn-Helix, green), HAMP (Histidine kinases, Adenyl cyclases, Methyl-accepting proteins and Phosphatases), sensor (orange) and transmembrane domain (blue) are depicted (*Lassak et al., 2013*). Saci_ArnR1 phosphorylation sites Tyr217 and Thr222 as identified in vivo are depicted in pink (*Reimann et al., 2013*) and Sso ArnR phosphorylation sites Tyr154 and Tyr155 are depicted in yellow (*Esser et al., 2012*). (B and C) In vitro phosphorylation assay. one μM ArnC (B) or ArnD (C) was incubated with μM ArnR or ArnR1 and γ³²-p-ATP and phosphotransfer from ArnC or ArnD to ArnR and ArnR1 was monitored, respectively. A representative phosphoimage of three independent experiments is shown.

(Fig. 4B). In experiments with *flaB* promoter fragments containing only ArnR box-1 or box-2, an increase in fluorescence signals was not observed (Fig. 4B), demonstrating that for efficient binding both boxes are required.

## ArnR and ArnR1 are involved in cross-regulation of various type-IV-pili structures

A cross-regulation between Aap-pili and the archaellum was proposed recently, as a deletion mutant of the central membrane protein-coding gene *aapF* of the Aap-system

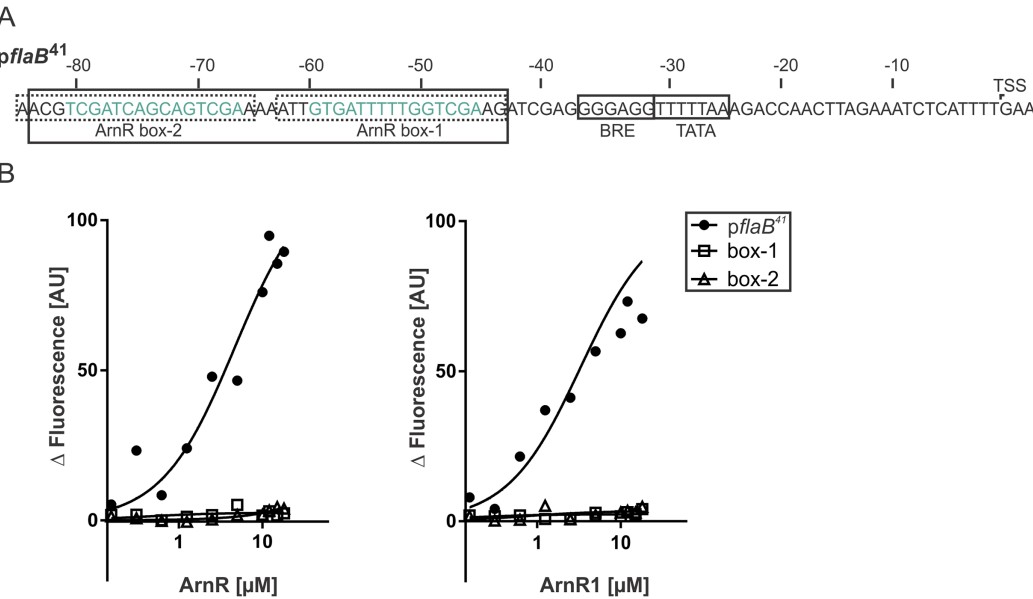

**Figure 4 Both ArnR boxes are required to facilitate ArnR and ArnR1 binding to p*flaB*.** (A) *flaB* (upper) promoter sequence. Numbers correspond to the distance respective to the transcription start site (TSS). The position of core promoter elements and cis-regulatory sequences is depicted in boxes (*Lassak et al., 2013*) BRE: factor B recognition element, TATA: TATA-box. p*flaB*$^{41}$ (black box) comprises the region of −43 to −84 and includes both ArnR boxes. The 15 bp regulatory elements ArnR box-1 and box-2 are highlighted in green and fragments used in this assay covering the sequence of either box are circled by dashed lines. (B) Net fluorescence obtained in microscale thermophoresis (MST) assays with increasing amounts of ArnR (left graph) and ArnR1 (right graph). 0.156–18 μM ArnR or ArnR1 were titrated in 2.5 nM Alexa Fluor 647 labeled DNA and subjected to thermophoresis. Curves were obtained from two independent experiments performed in duplicate. Fluorescence was normalized (ΔF) by subtracting the lowest measured fluorescence value determined within each experiment from all measured fluorescence values. Obtained curves were fitted using non-linear regression, with one site specific binding according to (y = F$_{max}$*X/(K$_D$ + X).

lead to hypermotile cells (*Henche et al., 2012a*). To analyze if apart from the archaellum ArnR and ArnR1 regulate the Aap- or the Ups-system of *S. acidocaldarius*, fragments of the *aapF* and *upsX* promoter sequences were subjected to MST analysis (Fig. 5A). Strikingly, ArnR bound to p*aapF* (Kd 1.5 μM) and p*upsX* (Kd 1.78 μM) (Fig. 5B, left graph), whereas ArnR1 did not (Fig. 5B, right graph). The promoter fragment of *saci_2122* (encoding a putative sugar binding protein) was used as negative control. Neither ArnR nor ArnR1 showed binding to p*Saci_2122*, supporting the idea that ArnR and ArnR1 exclusively regulate T4P related promoters. Taken together these results suggest that ArnR is involved in regulation of the archaellum, the Aap- and Ups-pilus of *S. acidocaldarius* while ArnR1 exclusively exerts its function in the archaellum.

Subsequently, we aimed to understand if there is an effect of phosphorylation of ArnR and ArnR1 on binding to p*flaB*, p*flaB* ArnR box-1, p*flaB* ArnR box-2, p*upsX* and p*aapF*. Therefore, ArnR and ArnR1 were incubated with ArnC and subjected to MST analysis. However, ArnR as well as ArnR1 was aggregating during the experiment, which hindered further analysis.

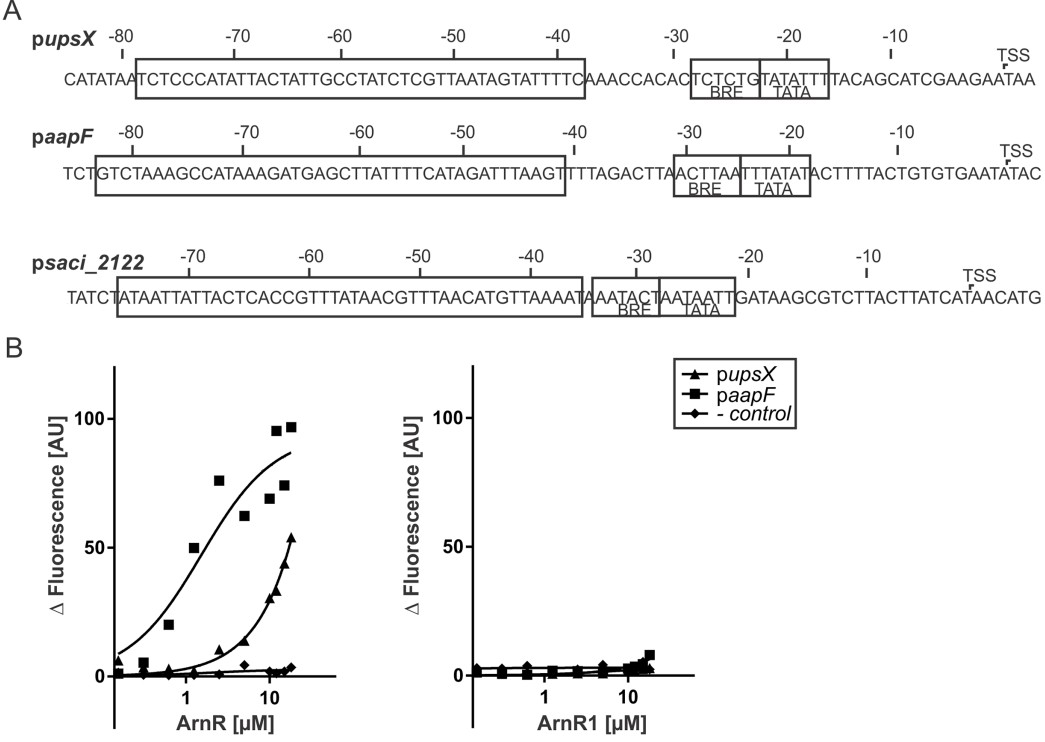

**Figure 5 ArnR binds to promoter sequences of other T4P structures.** (A) *upsX*, *aapF* and pSaci_2122 (- control) promoter sequences. Numbers correspond to the distance respective to the transcription start site (TSS). The positions of core promoter elements are depicted in boxes. BRE = factor B recognition element, TATA = TATA-box. The p*upsX* fragment comprises the region of −38−79 (black box). The p*aapF* fragment harbors the region of −41 to −82. (B) Net fluorescence obtained in microscale thermophoresis (MST) assays with increasing amounts of ArnR (left) and ArnR1 (right). 0.156–18 μM of protein was titrated in 2.5 nM Alexa Fluor 647 labeled DNA and subjected to thermophoresis. Curves were obtained from two independent experiments performed in duplicate. Fluorescence was normalized by subtracting the lowest measured fluorescence value determined within each experiment from all measured fluorescence values (ΔF). Obtained curves were fitted using non-linear regression, with one site specific binding according to (y = $F_{max}$*X/($K_D$ + X)).

**The deletion of *arnR* and *arnR1* affects transcription of *aapF* and *upsX***

It was recently shown, that the deletion of *arnR* and *arnR1* negatively affects the transcription rate of *flaB* and reduces motility of cells (*Lassak et al., 2013*). Since apart from binding to p*flaB* ArnR also bound to p*aapF* and p*upsX* in our MST analysis, the effect of *arnR* deletion on *aapF* and *upsX* levels was investigated in vivo. Up to now, the only known condition under which transcription of *arnR* is induced is nutrient depletion (*Lassak et al., 2013*). Therefore, a deletion mutant of *arnR* (*S. acidocaldarius* MW328) was subjected to nutrient depletion for 2 h and levels of *aapF* and *upsX* transcription were monitored relative to their transcription before starvation. The deletion mutant of *arnR1* was also assayed to analyze whether the deletion of *arnR1* affects transcription levels of *aapF* and *upsX* even though binding to their respective promoter regions was not detected in vitro. As depicted in Fig. 6, a log2 four-fold induction of *aapF* transcription after 2 h of nutrient limitation was detected in *S. acidocaldarius* MW001 (wild type) cells. Remarkably, *aapF* transcription was log2 four-fold
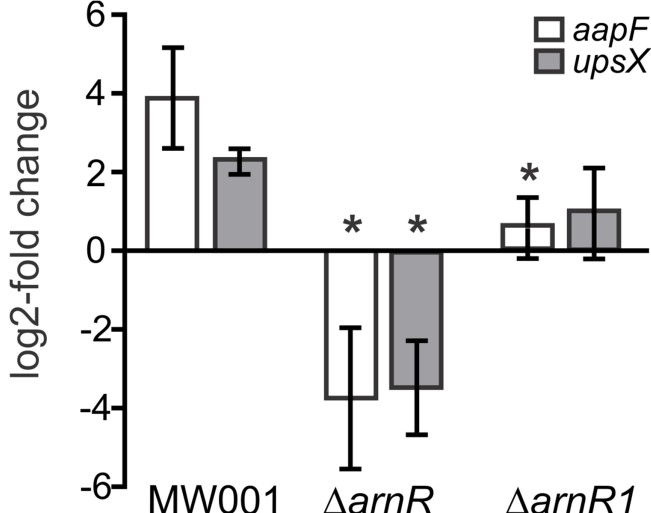

**Figure 6 Transcript levels of *aapF* (black) and *upsX* (gray) in *S.acidocaldarius* MW001(wild type) and *arnR* and *arnR1* deletion mutants.** Gene expression is depicted as log2-fold change after normalization to the housekeeping gene alcohol dehydrogenase (*adh*, Saci_1690). Data were obtained from biological triplicates analyzed in duplicate and are given as means ±SD. Values that are significantly different compared to the WT (Student's t-test, $p \leq 0.05$) are indicated by asterisks. *p*-value *aapF* transcript levels in Δ*arnR*: 0.0031; *p*-value *aapF* transcript levels in Δ*arnR1*: 0.00004.8, *p*-value *upsX* transcript levels in Δ*arnR*: 0.00013, *p*-value *upsX* transcript level in Δ*arnR1*: 0.06.

downregulated in the *arnR* deletion mutant and also significantly (>0.05) reduced in the *arnR1* deletion strain (Fig. 6). In *S. acidocaldarius* MW001 (wild type) a log2 two-fold induction of *upsX* was detected upon starvation, and the deletion of *arnR* (*S. acidocaldarius* MW328) led to a log2 four-fold downregulation of *upsX* (Fig. 6). No significant (>0.05) change of *upsX* transcript levels was found in the *arnR1* deletion mutant (Fig. 6). This confirms our MST binding assays and shows that ArnR is involved in the regulation of the *aap* and *ups* operons. Further, as identified by qRT-PCR ArnR1 is involved in the regulation of the *aap* operon.

## DISCUSSION

We investigated the function of ArnR and ArnR1 as transcription regulators of the archaellum, Aap and Ups pili system of *S. acidocaldarius*. Both proteins were purified to homogeneity and characterized in vitro. This is, to the best of our knowledge, the first biochemical analysis of membrane-bound transcription factors in archaea. A well-studied group of archaeal transcription regulators are the HTH domain containing Lrp-type transcriptional regulators. These regulate the amino acid metabolism and therefore bind small receptor molecules with their C-terminal RAM-domain (Regulation of Amino acid Metabolism) (*Peeters & Charlier, 2010*). They are functional in an oligomeric state and the minimal unit required for the interaction with the DNA is the dimeric species while higher multimers are often found in solution (*Leonard et al., 2001; Yokoyama et al., 2006; Peeters et al., 2007, 2009*). Dimeric Lrp-type regulators in archaea recognize 13–17 bp DNA-binding sites that contain an imperfect inverted repeat with an AT-rich

center (*Ouhammouch & Geiduschek, 2001*, *2005*; *Peeters et al., 2007*; *Peeters & Charlier, 2010*), as was also found in the *flaB* promoter sequence (*Lassak et al., 2013*). Lrp-type transcription regulators are well-known to form different oligomeric species. Based on these findings, a regulatory mechanism in which the different regulators regulate a variety of genes by varying their assembly form and also combining other transcription regulators into the assembly forms in response to environmental changes was proposed (*Koike et al., 2004*). The transition between different forms is thought to determine their binding specificity and ligands are thought to stabilize different assemblies and hereby allow the organism to adjust its overall transcriptome to environmental changes (*Koike et al., 2004*). We found oligomeric assemblies of ArnR and ArnR1 in this study. Therefore, such a mechanism might also underlie ArnR and ArnR1 function and allow regulatory effects on different promoters.

In the euryarchaeon *Methanococcus maripaludis*, the transcriptional activator EarA, which does not share homology with ArnR, promotes *fla* operon transcription by binding to a six bp consensus sequence (TACATA) that is present four times in the *fla* operon (*Ding et al., 2016*). Transcription of the *fla* operon in *Methanococcus maripaludis* is abolished upon elimination of the four EarA binding sites from the *fla* promoter (*Ding et al., 2016*). As mentioned earlier, the proposed target of ArnR and ArnR1 are two inverted repeats in the *flaB* promoter sequence (ArnR box-1 and box-2) and the lack of both boxes as well as mutation of box-2 significantly reduces *flaB* promoter activity (*Lassak et al., 2013*). In the in vitro studies performed here, ArnR and ArnR1 could only bind to the DNA molecule when both boxes were present. Furthermore, ArnR bound to promoter sequences of *aapF* and *upsX*. Both fragments did not harbor the inverted repeats. Since neither ArnR nor ArnR1 bound to the promoter sequence of the non-T4P related gene *saci_2122* (encodes a putative sugar binding protein) it is tempting to speculate that ArnR and ArnR1 are specifically regulating T4P surface structures of *S. acidocaldarius*.

The findings obtained from MST experiments were confirmed in vivo using qRT-PCR. It was recently shown, that the deletion of *arnR* leads to a decrease of *flaB* transcript levels of around 80% (*Lassak et al., 2013*) in nutrient depleted cells. In addition, we observed here a log2 8-fold decrease of *aapF* and *upsX* transcript levels in the *arnR* deletion variant during nutrient starvation as compared to wild-type cells. Thus, ArnR functions as transcriptional activator not only of the archaellum but is also involved in cross-regulation of the Aap and Ups pili system of *S. acidocaldarius* during nutrient depletion. Apart from that, a decrease of *aapF* but not of *upsX* transcript levels was observed in the *arnR1* deletion mutant under starvation conditions, indicating a different regulatory function of ArnR1. It is likely that ArnR1 also regulates the *flaB* and *aapF* promoters under conditions other than starvation as transcription of *arnR1* is not induced by starvation (*Lassak et al., 2013*). For instance, in *Methanocaldococcus jannaschii* hydrogen limitation promotes archaella synthesis while in *Methanococcus maripaludis* a decrease of archaellum gene transcription in response to leucine starvation was described(*Mukhopadhyay, Johnson & Wolfe, 2000*; *Hendrickson et al., 2008*). It is known that under different stress sources, such as nutrient or phosphate limitation or

osmotic and pH stress, *S. acidocaldarius* regulates gene transcription and translation according to its needs (*Osorio & Jerez, 1996*; *Lassak et al., 2012*, *2013*; *Buetti-Dinh et al., 2016*). To date, neither *arnR* nor *arnR1* were regulated under conditions other than starvation (e.g pH and salt-stress,(*Buetti-Dinh et al., 2016*)), raising the question of which membrane-related signal both regulators sense and transduce.

A connection between the archaellum and T4P was described for *Methanococcus maripaludis*, where archaella together with pili are important for binding to abiotic surfaces (*Jarrell et al., 2011*). *S. acidocaldarius* is by now the only identified species that possesses the Aap-pilus, which is the most abundant surface structure during exponential growth and majorly important during biofilm formation, where the archaellum only plays a role in the release of cells from the biofilm (*Henche et al., 2012b*). An actual cross-regulation of the archaellum, Aap- and Ups pilus was proposed as the deletion of the membrane spanning core component of the Aap-pilus promoted archaella formation and upregulation of the archaellum operon (*Henche et al., 2012b*).

Recently, a high number of phosphorylated proteins was detected in *S. acidocaldarius* and *S. solfataricus*, including ArnR and ArnR1 (*Esser et al., 2012*; *Reimann et al., 2013*). In this study, we observed phosphorylation of both regulators by the eukaryotic-like protein kinase ArnC (but not by ArnD). Phosphorylation is well-known post-translational modification that is crucially involved in archaeal signal transduction (*Esser et al., 2016*). ArnR and ArnR1 are phosphorylated in vivo (*Esser et al., 2012*; *Reimann et al., 2013*) and, as shown in this study, both proteins are phosphorylated by the serine/threonine eukaryotic protein kinase ArnC. Thus, apart from the two Tyr residues identified in the *S. solfataricus* ArnR homolog Saci_ArnR potentially harbors additional Ser/Thr residues that are post-translationally phosphorylated. Reversible protein phosphorylation is known to regulate the activity of archaellum regulators (*Hoffmann et al., 2016*; *Haurat et al., 2017*; *Li et al., 2017*). Therefore, it is tempting to speculate that ArnR and ArnR1 are also regulated by phosphorylation. Potentially, the phosphorylation status of ArnR and ArnR1 might regulate their promoter affinities to different T4P promoter sequence to regulate the expression of the surface structures. Unfortunately, DNA-binding analysis (MST) with phosphorylated ArnR and ArnR1 were impaired by protein aggregation. ArnR and ArnR1 possess a sensing domain in close proximity to their transmembrane anchors. Therefore, a membrane-associated stimulus was proposed to activate ArnR and ArnR1 (*Lassak et al., 2013*). It was proposed that a membrane-bound sensor kinase, ArnS, paradoxically inhibits *arnR* transcription while promoting ArnR translation (*Haurat et al., 2017*). In general, transcriptional activators interact with the basal transcription factors (TBP and TFB in archaea) to enhance gene transcription. However, a concentration-dependent dual function as activator and repressor was also described for transcriptional regulators (*Peeters, Peixeiro & Sezonov, 2013*). ArnR and ArnR1 are activators of motility, however it cannot be excluded that they act as transcriptional repressors on other T4P structures and under other conditions than starvation. A current model of the archaellum regulatory network is shown in Fig. 7. In summary,

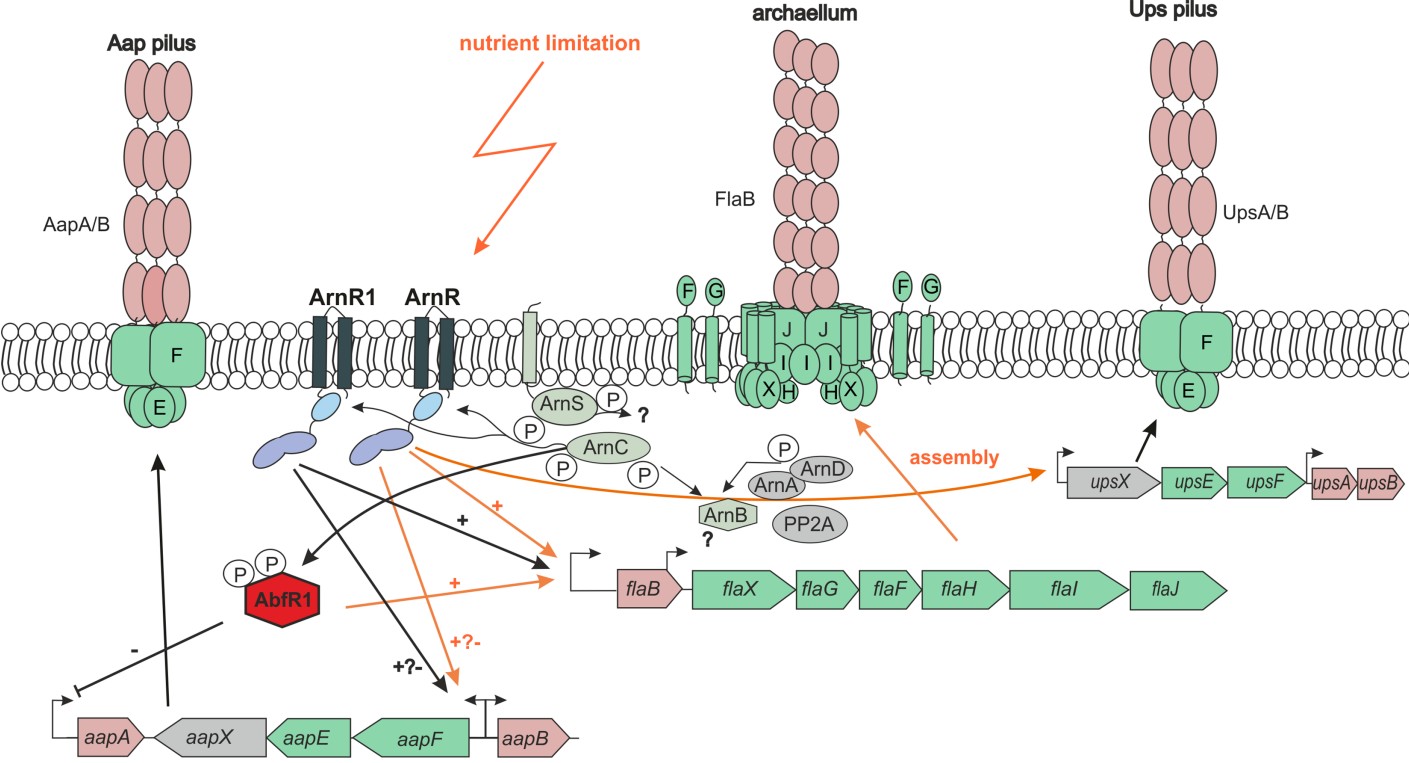

**Figure 7 Cross-regulatory network of T4P-like surface structures of *S.acidocaldarius*.** Operon and structural organization of Aap-pilus, archaellum and Ups-Pilus are shown. Known promoters of the operons are indicated by arrows. Orange arrows indicate pathways that are activated by nutrient limitation. Black arrows = unknown physiological conditions. − = repression, + = activation. P = phosphorylation. All by now identified key players of the archaellum regulatory network (Arn) and their regulatory activity on the Aap-, Ups-pili and archaellum is shown. ArnR is induced by nutrient limitation, and binds to *flaB*, *aapF* and *upsX* promoters. ArnR1 is induced by unknown physiological conditions and binds to *aapF* and *flaB* promoters. Both proteins are phosphorylated by ArnC. The membrane-bound sensor kinase ArnS is induced by nutrient limitation, but targets of phosphorylation are so far unknown. AbfR1, a repressor of *aapA* and activator of *flaB* is also phosphorylated by ArnC (*Orell et al., 2013*; *Li et al., 2017*). ArnA and ArnB are archaellum repressors under nutrient rich conditions, and are phosphorylated by ArnC and ArnD (*Reimann et al., 2012*). The phosphatase PP2A is involved in regulating the phosphorylation status of ArnA and ArnB (*Reimann et al., 2013*).

this study provides further evidence for cross-regulation of the archaeal motility structure and other T4P surface structures in *S. acidocaldarius*.

## CONCLUSIONS

Our results show that ArnR and ArnR1 can be purified to homogeneity, and form multimers.

Both proteins are phosphorylated by ArnC, but not ArnD, and thus it appears like phosphorylation is an additional regulatory level that is important for ArnR and ArnR1 function.

ArnR and ArnR1 require the presence of both Arn boxes of the *flaB* promoter as regulatory elements to bind to the promoter region. Further, ArnR but not ArnR1 bound to p*aapF* and p*upsX,* as was confirmed in vivo, where decreased *aapF* transcript levels were also detected in the *arnR1* deletion mutant. All in all, our data suggest that ArnR and ArnR1 are part of a regulatory network, that tightly regulates the expression of type-IV-pilus surface structures of *S. acidocaldarius*.

# ACKNOWLEDGEMENTS

We thank Prof. Carola Hunte for giving access to Nanotemper Monolith NT.115 Instrument. We thank Marleen van Wolferen and Chris van der Does for critical reading of the manuscript.

### Funding

Lisa Franziska Bischof and Maria Florencia Haurat were supported by the German research council in frame of the CRC 746. Lisa Franziska Bischof received additional funding from the Excellence Initiative of the German Research Foundation (GSC-4, Spemann Graduate School). The article processing charge was funded by the German Research Foundation (DFG) and the University of Freiburg in the funding programme Open Access Publishing. There was no additional external funding received for this study. The funders had no role in study design, data collection and analysis, decision to publish, or preparation of the manuscript.

### Grant Disclosures

The following grant information was disclosed by the authors:
German research council in frame of the CRC 746.
Excellence Initiative of the German Research Foundation (GSC-4, Spemann Graduate School).
German Research Foundation (DFG) and the University of Freiburg in the funding programme Open Access Publishing.

### Competing Interests

Sonja-Verena Albers is an Academic Editor for PeerJ.

### Author Contributions

- Lisa Franziska Bischof conceived and designed the experiments, performed the experiments, analyzed the data, prepared figures and/or tables, authored or reviewed drafts of the paper, approved the final draft.
- Maria Florencia Haurat conceived and designed the experiments, approved the final draft.
- Sonja-Verena Albers conceived and designed the experiments, contributed reagents/materials/analysis tools, prepared figures and/or tables, authored or reviewed drafts of the paper, approved the final draft.

### Data Availability

 The raw data are available in a Supplemental File.

### Supplemental Information

Supplemental information for this article can be found online at http://dx.doi.org/10.7717/peerj.6459#supplemental-information.

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
