# Peer review of "Two membrane-bound transcription factors regulate expression of various type-IV-pili surface structures in Sulfolobus acidocaldarius"

_PeerJ, doi:10.7717/peerj.6459_

## Round 0.1 · original submission · Minor Revisions

I concur with the many suggested corrections to the manuscript by both reviewers. Please attend to these as best you can. Sincerely,

--Craig Moyer

·

Basic reporting

See general comments

Experimental design

See general comments

Validity of the findings

See general comments

Additional comments

This study deals with two transcriptional regulators, ArnR and ArnR1 from the thermoacidophilic archaeon Sulfolobus acidocaldarius. They have been known to be involved in the transcriptional regulation of the archaellum operon by targeting the promoter of flaB that encodes the archaellum filament protein. ArnR and ArnR1 are capable of forming oligomers and are phosphorylated by the Ser/Thr kinase ArnC. Apart from binding to pflaB, ArnR but not ArnR1 bound to promoter sequences of aapF and upsX, which encode components of the adhesive (aap) and UV inducible pili (ups), demonstrating a regulatory connection between different surface appendages of S. acidocaldarius. The information provided is new, especially the fact that ArnR may be involved in the regulation of other surface appendages. The manuscript however needs some editing. In particular, the discussion should be trimmed and refined/clarified for the general reader. Perhaps a figure of the regulation network would make it much easier to understand. The conclusion section is not necessary unless this is the format of the journal. A list of specific comments, both major and minor is shown below.

2nd affiliation needs an address. Spemann Graduate School of Biology and Medicine (SGBM)

However, sigma factors are absent from archaea raising the question of how gene transcription is regulated. This sentence seems strange. If Archaea were to use bacterial transcriptional machinery, the absence of sigma factors would be intriguing, but this is not the case. I do not think the authors have to refer to the absence of sigma factors in order to raise the question as to how gene transcription is regulated.

While ArnR is conserved in Sulfolobales and Desulfurococcales, the ArnR paralog ArnR1 is exclusively found in S. acidocaldarius (Lassak et al., 2013). Is the distribution still the case taking into account the dramatic increase in genome sequences in recent years? I ask this as the citation is from five years ago.

Lines 95 and 101: Organisms in italic or is this the journal style for subtitles?

Line 147: DNA in italic?

Line 181: This was surprising. N-terminal His10-tag under control of an arabinose inducible promoter. How is the protein integrated into the membrane? Is there not a signal sequence at the N-terminus? A comment would help.
Related to this, although archaeal membrane proteins such as SecYEG have been known to assemble in bacterial membranes, is it not worth noting? Is the insertion machinery analogous between archaea and bacteria? Is the majority of protein localized in the membrane fraction? Some comments would help.

Change y32 to gamma (greek letter)32

Description of Figure 3 text lines 205-214: Why do the authors not mention the self phosphorylation of ArnD? Understanding the meaning of the bands takes a while and it would be helpful if this were written here.

Line 237: the first p in roman? Are paapF and pupsX designated somewhere?

Are there any comments or discussion on the quite large Kd values?

Line 262: mutant in italic?

since neither ArnR nor ArnR1 bound to the promoter sequence of the non-T4P related gene saci_2122 (encodes a putative sugar binding protein) it is tempting to speculate that rather the length of the DNA fragment and/or distance to core promoter elements such as the BRE, TATA-box and/or transcription start site are important for ArnR and ArnR1 function in regulating T4P surface structures of S. acidocaldarius. The logic is hard to understand here, please clarify.

Line 324: jannaschii

Line 325: M. maripaludis

Line 331: Is this e.g.?

Line 338: S. acidocaldariusis

Line 345: S. acidocaldarius in italic

Line 364: What exactly tempts the authors to speculate that phosphorylation of ArnR and ArnR1 defines their specificity to different T4P promoter sequences?

Line 372: ArnR/1?

·

Basic reporting

There were various inconsistencies throughout article that should be corrected prior to publication:

Line 70 – Spell out HAMP
Line 110 – Should list components of PBST
Line 149 – 16,000 instead of 16.000 rpm
Line 166 – “and” does not need to be italicized
Line 170 – aapf should be aapF
Line 171 – change “log2-folg” to “log2-fold”
Results
Figure 2 caption; second line - list separately as ArnR/ArnR1 for consistency
Lines 193 – 214 – The beginning of this paragraph contains a lot of information that seems like it would fit better in the Discussion section.
Figure 4 caption, fourth line – change pFlaB41 to pflaB41
Lines 237/238 – change paapF to paapF to stay consistent with italization between paapF and pupsX
Line 246 – change Arnr1 to ArnR1
Figure 6 caption, second line – italicize adh
Figure 6 caption, last line – delta symbol needs to be added instead of DarnR and darnR1.
Line 293 - list separately as ArnR/ArnR1 for consistency
Line 308 – add bp after 40
Line 345 – italicize S. acidocaldarius
Line 351 – change Arnr1 to ArnR1
Line 352 – remove “by on” replace maybe with “at”
Line 372 – change Arnr1 to ArnR1
Line 386 – change hosphorylation to phosphorylation

Experimental design

I think that the experimental design for the experiment were adequate, but a few sections in the Materials and Methods section need work. First, the phosphorylation studies were not even mentioned in the Materials and Methods. This should be added prior to publication of the article. This would also help in understanding Figures 3B and 3C which took a while to wade through because of the lack of information in M & M. Obviously not enough information to replicate this specific set of experiments.

The second issue was related to the qRT-PCR section in the M & M:

Lines 159 – 172 – qRT-PCR Section of Methods is confusing. On lines 165 and 166 the authors indicate they are targeting flab and arnR/arnR1, while in lines 169 and 170 they talk about normalization of expression of aapf and upsX. Since data was only included for aapf and upsX, I can assume that lines 165 and 166 should be corrected. They also listed Table S2 as listing primers; however, primers are listed in Table S3. Listing of primers in Table S3 is also not consistent. upsX primers are listed as qPCR primers, while the aapF and adh are listed as qRT-PCR primers. Italics are also used inconsistently in table. Please re-write section so that it is consistent with research task performed. Would also be worthwhile to state why adh was used as housekeeping gene.

Validity of the findings

The research was impactful and novel, showing that ArnR and ArnR are part of the regulatory network of S. acidocaldarius for activation of motility. Likewise, the work showed that there was regulation of other motility and surface structures.

The data presented is robust and the raw data and statistical analysis was sound and well controlled.

The conclusions of the research are succinctly stated and linked to the original question discussed in the introduction.

---

## Round 0.2 · accepted · Accept

Reviewer's comment were adequately addressed.

#